# Potential of *Artemisia dubia* Wall Biomass for Natural Crop Protection

**DOI:** 10.3390/plants12213750

**Published:** 2023-11-02

**Authors:** Aušra Bakšinskaitė, Vita Tilvikiene, Karolina Barčauskaitė, Dalia Feizienė

**Affiliations:** Lithuanian Research Centre for Agriculture and Forestry, Instituto av. 1, LT-58344 Akademija, Lithuania; vita.tilvikiene@lammc.lt (V.T.); karolina.barcauskaite@lammc.lt (K.B.); dalia.feiziene@lammc.lt (D.F.)

**Keywords:** *Artemisia dubia* Wall, biomass, allelopathy, phenolic compounds

## Abstract

The Green Deal strategy has the very ambitious goal of transforming the European Union into the first climate-neutral continent by 2050. For the agricultural sector, one of the main challenges is to reduce the use of synthetic fertilizers and pesticides. Crop protection measures aim to maintain and ensure certain standards of yield and quality, which are generally achieved by the control of pests, diseases, and weeds. One of the possibilities to reduce the use of pesticides could be allelopathic plants, which are not only potential sources of allelochemicals but also renewable biomass sources. The aim of this study was to analyze the productivity of *Artemisia dubia* Wall and evaluate the allelopathic effects of biomass on crops and weeds. It was determined that the biomass productivity of *A. dubia* varied from 2 to 18 t ha^−1^, depending on how many times it is cut during the growing season and the fertilizer rate. *A. dubia* has allelopathic properties, which were verified using an aqueous extract and can completely suppress the germination of *Taraxacum officinale* seeds. Young plants harvested in the middle of summer were characterized by the highest number of phenolic compounds. This shows the strong allelopathic effect of *A. dubia* biomass on other plants.

## 1. Introduction

The use of synthetic pesticides and fertilizers is one of the key factors for growing quality crops. Nevertheless, the increasing use of synthetic additives has negative effects on the environment: it accelerates climate change and reduces soil fertility. For these reasons, in 2019, the European Commission launched a new Green Deal strategy. One of the main challenges will be reducing synthetic fertilizers and pesticides by 50% in the agricultural sector. To achieve this goal by 2030, it is necessary to look for alternatives to synthetic chemicals that would help to implement this plan [1]. It is expected that plants, especially allelopathic ones, could be potential sources of allelochemicals used for crop protection and at the same time effectively utilized as renewable biomass.

Allelopathy is the effect of plants on other plants when biochemical substances are released into the environment, which together directly or indirectly affect the growth of neighboring plants or microorganisms in a certain natural or agricultural system [2]. Interactions between organisms can be greatly influenced or even changed by biotic and abiotic factors, sunlight, wind, soil and air moisture, temperature regimes, and community structure [1]. Allelopathy plays an important role in agroecosystems, causing significant interactions between crops, crops, and weeds [3]. These interactions between plants have great potential for improving plant productivity, maintaining high genetic diversity, and managing weed and pest diversity [4]. Allelopathic properties can be found in various parts of plants, from greens and flowers to roots, bark, soil, or even mulch. Most allelopathic plants store defense chemicals in their leaves and accumulate the highest amount in autumn. When the leaves fall to the ground and decompose, these toxins have the opportunity to affect nearby plants [5].

Excreted substances that exhibit allelopathic properties are called allelochemical compounds [4]. These compounds can disrupt developing plant roots and seedlings and cause several other damages to plants [6]. Polyphenols are the most abundant and widespread group of bioactive molecules [7]. Depending on the strength of the phenolic ring, polyphenols can be divided into many classes, but the main classes of polyphenols are phenolic acids, flavonoids, stilbenes, phenolic alcohols, and lignans. It has also been observed that plants that release higher amounts of chemicals often have a negative effect on other living organisms in the environment [3]. Higher concentrations of allelochemicals may significantly inhibit plant growth, while low concentrations can have a positive effect on plant physiological processes [8].

By secreting physiologically active substances, plants are able to compete well with each other. As a result, allelochemicals can be used as natural plant protection products [9]. A. Reiss and other researchers pointed out that there is a large contribution of allelopathy to the overall result of weed control, and it has been shown that allelopathy can contribute significantly to weed control in crops [10]. Therefore, the effects of weeds on crop production, the challenges of weed management, potential allelopathic plants, and the use of allelopathy for weed control are receiving increasing attention [6,11].

Nevertheless, the variations of plants that are rich in allelochemicals are mainly grown in warm climate conditions. But the ones that are very promising in warm climate countries cannot be grown in a northern climate. Therefore, the northern part of the temperate climate zone needs to look for alternative crops, which could be productive and rich enough in active compounds as well as tolerant to frost. *Artemisia* is one of the diverse generals of the *Asteraceae* family with very important alchemical, secondary metabolites [12]. The genus *Artemisia* includes more than 500 species and is widespread in Asia, Europe, and North America [13]. Plants of this genus are mostly used in medicine but when studying their chemical composition, it becomes possible to use them in different fields. Under the influence of plants with a wide range of applications, they contain essential oil components that have insecticidal, antimicrobial, and antiparasitic properties [13,14]. Plants of the *Asteraceae* family are rich in biologically active compounds such as polyphenols, terpenes, and flavonoids.

*Artemisia* plants can be 1 to 2.5 m tall. The aerial parts and seeds of these plants are high-quality fodder and high-calorie biofuels [15]. Naturally growing perennials are not often studied and do not yet have a special purpose. These plants are able to adapt well enough to climatic conditions; they are able to grow new, strong, and fast-growing stems every year. In Lithuania, there are 11 species of this *Artemisia* genus, of which the most popular and most researched are *A. abrotanum*, *A. absinthium*, *A. dracunculus*, *A. vulgaris,* and the little-known *A. dubia* Wall. [14].

Several species of the *Asteraceae* family are known to have allelopathic compounds that can reduce seed germination and sprouting of other plants [16]. On the other hand, *Artemisia dubia* Wall is little-known in Lithuania and other countries. In terms of productivity, *Artemisia dubia* Wall surpasses *Artemisia vulgaris* L. [17] and can produce a considerable amount of biomass, from 3.0 to 27.6 t·ha^−1^ [18,19]. They are mostly found in the East Asian region and are commonly grown as ornamental plants in Europe [20,21]. In northern climates, *A. dubia* begins to grow in May and grows intensively in June. Flowering begins in August [22], but because of the northern climate, it does not produce seeds. At the end of the growing season, the canopies are dense and have thick stems. According to A. Kryževičienė and other scientists, *Artemisia dubia* can be grown in various conditions [22], such as on abandoned and less suitable lands for nutritional fodder, and there is an opportunity to introduce long-term *A. dubia* plantations and thus make proper use of all cultivated lands [14].

It is expected that allelopathic plants, such as *Artemisia dubia,* could be widely used in sustainable and organic agriculture because of their potential role not only in controlling weeds and pests but also in promoting crop growth. Nevertheless, there is little information about their growth potential, chemical composition, and effect on plants and the soil. Research hypothesis that different density, fertilization rates, and biomass harvesting time influences the amount of phenolic compound formation, which is important for plant protection. The aim of this study was to analyze the productivity of *Artemisia dubia* Wall and evaluate the allelopathic effects of biomass on crops and weed seeds. It is expected that *A. Dubia* biomass, which has allelopathic properties, can be an alternative to synthetic pesticides. After analyzing the obtained results, it will be possible to determine the use of *A. dubia* biomass as plant protection, which is also a renewable resource. This will not only provide an opportunity to reduce the use of synthetic pesticides but will also be sustainable.

## 2. Results

### 2.1. Productivity

The results of four years of experiments show that the biomass productivity of *Artemisia dubia* Wall was the highest in crops fertilized with 180 kg ha^−1^ of ammonium nitrate and cut once per vegetation season (Figure 1). In treatments with lower crop density (5000 plants ha^−1^), the productivity of dry biomass decreased each year, and only the yield of the cuts in harvest III per growing season did not differ significantly in all four years of cultivation, except for cases where plants were fertilized with 180 kg ha^−1^ of nitrogen fertilizers. Nevertheless, in 2021, *Artemisia dubia* harvest clearly stood out when fertilizing 90 kg per ha and 180 kg per ha. This could have been influenced by the high temperature and low rainfall in the summer of 2021.

A similar trend remained in denser treatments (20,000 plants ha^−1^). As in rare crops, the highest yield of *Artemisia dubia* Wall was when fertilized with 180 kg ha^−1^ of nitrogen fertilizer and cut once per growing season.

At the beginning of the vegetation, the germination of *A. dubia* Wall did not differ significantly and ranged from 62.7 to 82.7 plants m^−2^. After evaluating the number of plants one month after spring fertilization, the number of plants increased to 112.0 units m^−2^; however, no significant differences were found between different treatments. The number of plants per cut was influenced by the fact that *A. dubia* Wall was cut only once per growing season, which did not change the number of plants. Nevertheless, when assessing the number of *A. dubia* Wall, a reduction in plant stems was observed in all fields from one cut until harvest during the growing season, and there were no significant differences between different fertilizations. The decline of *Artemisia dubia* during the growing season is also influenced by the competition between the plants themselves. When plants grow, they need nutrients, moisture, and light, and as a result, plants begin to compete not only with weeds but also with plants of the same species. Stronger and larger plants outgrow other, less resistant, and weaker plants, overshadowing them. It can be observed that the number of plants was lower in the N_90_ fertilized plots in both harvest I and harvest III. This may have been due to higher weed content in these fields. Weeds could overshadow them by competing with young *A. dubia* Wall seedlings.

*Taraxacum officinale* and *Elytrigia repens* were the most dominant weeds in the field. Comparing the fields of harvest I during the growing season with harvests II and III, it was observed that the population of *Taraxacum officinale* is significantly lower. In the fields of one harvest, the abundance of *Taraxacum officinale* stood out the most in the unfertilized field, which contained from 9.3 to 18.7 units m^−2^. In fertilized N_90_ fields, these weeds were less than from 2.7 to 13.3 m^−2^, except for the first count, 18.7 units m^−2^, when the stalks were still small. After fertilizing them, the dandelions made better use of the spread fertilizer. Midway through the *A. dubia* Wall vegetation, weed levels remained similar. This shows that the plants are completely overshadowing *Taraxacum officinale*. Another one of the most dominant weeds was *Elytrigia repens*. In the first harvest during the growing season, *Elytrigia repens* was more abundant in fields with N_180_ fertilization compared to less fertilization with N_90_ or no fertilization with N_0_. *Elytrigia repens,* which had the most N_180_ fertilization—54.7 units m^−2^. During the next weed count, the amount of *Elytrigia repens* decreased from 36.0 to 6.7 units m^−2^. These weeds increased in population after each mowing but decreased again as *A. dubia* Wall grew.

In addition to these two discussed weeds, which were the most prevalent in the fields of the field experiment, *Sonchus arvensis* and *Cerastium arvense* also spread and persisted until the end of the growing season.

Single common weeds included *Lamium purpureum*, *Rumex crispus*, *Deschampsia cespitosa*, *Galium aparine*, *Stellaria media*, *Veronica arvensis*, *Glechoma hederacea,* and *Viola arvensis*. All these detected weeds were most prevalent in the fields of harvest III during the growing season. In the majority of all the weeds found, only *A. dubia* Wall was present at the beginning of the growing season or after harvesting. This means that *A. dubia* Wall outgrows the smaller weeds, blocking them from light and absorbing nutrients.

### 2.2. Allelopathic Influence on Crop and Weed Seed Germination

An allelopathy study was conducted to analyze the effect of *Artemisia dubia* Wall biomass extract on the germination of seeds. Analyzing the obtained data on the effect of *A. dubia* aqueous extracts of different concentrations on the germination of spring wheat seeds, it was found that spring wheat germinated 99.17% in distilled water (Figure 2). A significant difference was in the aqueous extract of *A. dubia* with the strongest concentration of 1:10, which inhibited the germination of spring wheat seeds by 8.4%. In the extracts of lower concentrations of *A. dubia*, the germination of spring wheat seeds was inhibited by 1.68–5.89% compared to distilled water, but no significant differences were obtained.

The germination of spring rapeseeds was mostly influenced by the highest concentration (1:10) of aqueous extracts where rapeseed did not germinate at all (Figure 2). Extracts of a concentration of 1:50 did not significantly reduce the germination of spring rapeseeds compared to distilled water. Most spring rapeseeds germinated in aqueous extracts of lower concentrations of 1:1250 and 1:6250 were, respectively, 94.17 and 93.33%. It can be stated that in the passages with the lowest concentrations, the germination of rapeseed was slightly higher by 1.8–2.7%. When comparing the influence of aqueous extracts of *Artemisia dubia* Wall at concentrations of 1:50 and 1:6250 on the germination of rapeseeds, it was found that germination significantly increased by 17.7%.

Analyzing weed seed germination in different concentrations of *Artemisia dubia* Wall aqueous extracts had the highest effect on *Taraxacum officinale* seed germination compared to seed germination of *Chenopodium album*. The germination of *Chenopodium album* seeds in distilled water and different concentrations of *Artemisia dubia* Wall aqueous extracts had no significant effect, except for at a concentration of 1:6250 (Figure 2). The germination of *Chenopodium album* seeds was significantly promoted by the aqueous extract at a concentration of 1:6250 (20.9%) compared to distilled water. In distilled water and in the strongest concentration (1:10) in the aqueous extract of *Artemisia dubia* Wall, there are no differences between the germination of *Chenopodium album* seeds, as in both cases germination reached 71.67%.

The conducted research found that *Taraxacum officinale* seeds did not germinate in the aqueous extract with the strongest concentration (Figure 2). At lower concentrations (1:250, 1:1250, and 1:6250) of aqueous extracts, *Artemisia dubia* Wall inhibited the germination of *Taraxacum officinale* from 1.7 to 3.5% compared to distilled water. The germination of *Taraxacum officinale* was most inhibited by aqueous extracts of *Artemisia dubia* Wall at a concentration of 1:50, and germination was significantly smaller by 45.6%.

### 2.3. Effect of Artemisia dubia Wall Extracts on Plantlet Growth

The height of shoots and the length of roots were evaluated in the study. Figure 3 presents the difference between distilled water and *A. dubia* aqueous extracts of different concentrations. The height of spring wheat shoots at a concentration of 1:250 was the most distinguished, and this concentration of the aqueous extract was significantly inhibited by 30% compared to distilled water. The same inhibition effect was also observed in the height of rapeseed shoots, such as *Artemisia dubia* biomass aqueous concentrations of 1:10 and 1:50. Extracts with concentrations of 1:1250 and 1:6250 did not have a significant difference in shoot heights; they ranged from 11.2 mm to 12.8 mm. By analyzing the diagram, it can be stated that the height of shoots was also inhibited by aqueous extracts of 1:50 and 1:1250 concentrations, but there were no significant differences. However, the height of rapeseed shoots differed the most in the weakest (1:6250) concentration, where the shoots were taller compared to distilled water. The extract concentration of 1:50 had the greatest effect on rapeseeds—they did not germinate.

During the germination of spring wheat seeds, their root length in distilled water reached 37.6 mm (Figure 3). Of all the aqueous extracts, the 1:1250 concentration stood out the most and had a 9.31% stimulating effect on root growth (41.1 mm). Aqueous extracts of *Artemisia dubia* Wall at concentrations of 1:10, 1:50, and 1:250 had a significant inhibitory effect on spring wheat root growth.

When analyzing the influence of different concentrations of aqueous extracts of *Artemisia dubia* Wall on rapeseed root length, a concentration of 1:1250 was found to stimulate rapeseed root growth by up to 35%, and a significant difference was found (Figure 3). A weaker concentration (1:6250) also stimulated rapeseed root growth, but the effect was slightly lower (10.7%). Extracts with concentrations of 1:50 and 1:250 were distinguished by their inhibitory effect on the length of the roots, and the length of the roots was significantly smaller.

The growth of *Chenopodium album* roots was stimulated by the 1:250 concentration of the aqueous extract of *A. dubia* Wall; they were significantly longer (19.2%) than in the extracts of other concentrations (Figure 4). Aqueous extracts with a concentration of 1:50 also had a stimulating effect; the length of the roots was 8.9 mm. The strongest concentrations of the aqueous extract (1:10) had the most inhibitory effect on root growth, up to 39.7%, compared to distilled water.

When analyzing the results of the conducted research, it can be seen that all concentrations of the aqueous extract of *A. dubia* Wall significantly inhibited the growth of *Taraxacum officinale* roots compared to distilled water—14.9 mm (Figure 4). However, it should be noted that in the strongest (1:10) concentration of aqueous extracts, *Taraxacum officinale* seeds did not germinate. Aqueous extracts with a concentration of 1:50 had an inhibitory effect on root length, which significantly inhibited 65.8% root growth compared to distilled water.

When analyzing the influence of aqueous extracts of *Artemisia dubia* Wall on the height of *Chenopodium album* shoots, it is observed that the strongest concentrations of aqueous extracts (1:10) completely inhibited the growth of seed sprouts, and shoots were not formed.

Aqueous extracts of *Artemisia dubia* Wall biomass affected the height of *Taraxacum officinale* shoots (Figure 4). The aqueous extract with a concentration of 1:1250 had no effect, and the height of the shoots was 2.0 mm, the same as in distilled water. Among all concentrations of aqueous extracts of *Artemisia dubia* Wall, the aqueous extract at a concentration of 1:250 had a weak inhibitory effect (5.0%). A significant difference was found in the aqueous extract of *Artemisia dubia* Wall at a concentration of 1:50, where the height of *Taraxacum album* shoots was 0.7 mm or 1.5 times shorter compared to distilled water. Shoot growth was also significantly inhibited (20%) by the weakest (1:6250) concentration of the aqueous extract of *Artemisia dubia* Wall.

### 2.4. Phenolic Compounds

The total number of phenolic compounds was determined in each of the treatments. The biomass taken during the second cut of the cuts of harvest III was characterized by the highest amounts of phenols. The highest amount (157 mg/g) was 5000 plants per hectare in the unfertilized treatment (Figure 5). Compared to more densely planted plants, the number of phenols decreased slightly (about 10%). When *A. dubia* Wall was fertilized with N 90 and 180 kg ha^−1^, the amount of phenols decreased by up to 20% compared to the unfertilized variant. Analyzing the diagram (Figure 5), it can be seen that the number of phenols is the highest in all unfertilized treatments.

The last analysis of the cut samples showed that mature plants do not contain a lot of phenols, and this is shown by the first harvest, second harvest, and third harvest. There is no significant difference between these treatments, and nitrogen content did not affect any of the treatments.

## 3. Discussion

This study investigated the crop biomass utilization potential of *Artemisia dubia* Wall. Nowadays, in the search for an alternative to synthetic pesticides, *A. dubia* Wall has the opportunity to be used as an alternative. For this, attention was paid to the productivity of these plants and the allelopathic influence of their properties on crop germination. These properties allow the application of natural resources as an alternative to synthetic pesticides. The overall goal was to use the results to adapt to further field experiments. The results are discussed below in relation to the objectives.

### 3.1. Biomass Yield

In the European Nordic–Baltic region, the species *Artemisia dubia* Wall is not as common as in Asian countries, where the chemical composition and utilization possibilities of biomass are widely studied. *Artemisia dubia* Wall is one of the most productive non-food perennial crops. The biomass yield per hectare can be equal to that of *Miscanthus giganteus*, *Sida hermaphrodite*, *Festuca arundinaceous* [19,23], *Urtica dioica* [24], and *Silphium perfoliatum* [25]. The dry matter yield of energy crops as reported by Jablonwski et al. [26] depends not only on soil fertility, rainfall, and climatic conditions but also on plant age and harvest time [26]. *A. dubia* can quickly spread throughout the world by vegetative propagation. However, in the nemoral climate zone, *Artemisia dubia* crops cannot produce seeds [14]. Due to its infertility, *A. dubia*, like *Arundo donax* [27] and *Artemisia annua* in spring, when the vegetation resumes, the rhizomes explore the soil to the sides and allow the plant to expand [28]. In this way, the plants renew themselves every year and can maintain a high yield, even for several years. This is evidenced by the diagram in Figure 1, which shows the productivity of *A. dubia* every year. However, regardless of the use of nitrogen fertilizers and the number of cuts during the growing season, the yield decreases every year.

Fertilization can increase biomass production [27]. The addition of nutrients promotes better rhizome development. With the emergence of new seedlings every year, it is possible to maintain a similar amount of biomass. However, it is difficult to maintain the productivity of plants every year. Reasons include different climatic conditions or the thinning of plants, which are influenced by two or three harvests per season. Frequently mowed *A. dubia* plants are unable to produce a mass that can suppress weeds during the short summer season. Therefore, competition occurs, and weaker plants (unfertilized) become unable to compete with weeds. In the graph analyzed in Figure 1, it is possible to see the difference in biomass during several harvests.

Nevertheless, the availability of nutrients not only increases the yield but also defines the qualitative improvement of the biomass. Fertilizing with higher amounts of nitrogen fertilizers results in stronger and bigger plants. Like other non-food plants, *A. dubia* can produce a considerable amount of biomass without fertilizing, so fertilizing is only recommended, but not necessary [29].

### 3.2. Effects of Aqueous Extracts on Crops and Weeds

Allelopathy is defined as one plant produing chemicals that affect the growth and development of other plants. Allelochemicals can promote or inhibit crop germination and shoot or root length. Jinxin Li and other researchers considered a potential strategy for the development of environmentally friendly bioherbicide [30]. Often high concentrations of allelochemicals can adversely affect other plants, and low concentrations can sometimes be stimulatory [31].

In the present study, it was found that there was different seed germination in distilled water and *A. dubia* aqueous extracts. The germination of seeds and the root and shoot length of all studied crops and weeds were inhibited by the aqueous extract of *A. dubia*. Among all concentrations of the aqueous extract of *A. dubia* seed germination, shoot and root lengths were inhibited the most at a 1:50 concentration. The inhibition of the strongest aqueous concentrations of *A. dubia* on the development of seedlings and the length of roots and stems is also observed by B. B. das Mallik [16].

Italian scientists found that plants of the genus *Artemisia* can stimulate the germination of carrots, cabbage, and rapeseed. However, plant extracts from different species of the genus can stimulate different plants. In contrast, the Italian scientist E. Panacci and his colleagues [32] found that the aqueous extract of *Artemisia vulgaris* promotes the germination of *Barssica napus* seeds. And, in our study, the aqueous extract of *Artemisia dubia* inhibited *Barssica napus* seed plants but promoted root growth. The biological activity of plants with allelochemicals depends on the concentration. Characteristic responses include stimulation at low concentrations of allelochemicals but inhibition as concentrations increase [33]. Therefore, it is concluded that its bio-stimulant effect on crops and herbicidal effect on weeds can make it an ideal solution in an integrated crop protection program. In order to suppress weeds and increase the competitiveness of cover crops, it is necessary to pay attention to the concentration of extracts [32].

### 3.3. The Importance of Biologically Active Substances

Phenolic compounds are secondary metabolites produced by higher crops, which play multiple essential roles in crop physiology. They are synthesized by plants in response to ecological and physiological conditions, mainly when they experience biotic or abiotic stress [34,35]. Also, phenols play an important role in protecting against aggression from plant pathogens and animal herbivores and respond to various abiotic stress conditions, such as precipitation and ultraviolet radiation [36]. However, the study was conducted to see which period *A. dubia* accumulates phenol the most. From the obtained results, it can be seen that the highest total amount of phenols occurs during the first and second harvests in June–July. Then, the green biomass of *A. dubia* is the richest in phenolic compounds, and this is also noticed by J.E. Alba-Mejia and other researchers [37]. Abiotic factors can influence the number of phenols in plants, such as rainfall that was higher than the long-term average rainfall in the harvest year. This shows that the environment can influence plant physiology.

## 4. Materials and Methods

Field and laboratory research of *Artemisia dubia* Wall were conducted at the Lithuanian Research Centre for Agriculture and Forestry (LAMMC), Institute of Agriculture, Kėdainiai district, Akademija (55°23′50″ N, 23°51′40″ E). Plants were growing on light loamy soil. Chemical characteristics of the soil were at 0–20 cm of depth. The soil agrochemical indicators determined in dry matter are as follows: pH_KCl_—6.56, N_sum_.—0.611%, K_2_O—0.300%, P_2_O_5_—0.075%, and C_org_.—7.93%.

### 4.1. Field Experiment

*Meteorological conditions.* Weather data were collected during 2019–2022 at the meteorological station located at the Akademija experimental sites (Figure 6). The temperature of the vegetation period (April–October) was higher than the average annual temperature during all field experiment years. Nevertheless, the highest temperature was recorded in July 2021. According to the weather data from the meteorological station, the amount of precipitation was distributed very unevenly in all years. The spring in 2019 and 2021 was dryer compared with other experiment period years. However, the summers of 2020 June and 2022 April–July were unusually wet during the growing season.

*Experimental treatments.* The field experiments were carried out in 2019–2022 May–October. The experiment was performed with 18 treatments. Two plant densities X 3 fertilization rates X 3 cut treatments = 18 plots. Each field was 14 m long and 3 m wide. The field experiment is affected by three factors (Table 1).

Plant fertilization was divided into two times, depending on the amount of nitrogen per ha. The first fertilization of 90 N kg ha^−1^ is carried out in the spring (in May) after the beginning of plant vegetation. During the second fertilization (in June), the remaining dose of nitrogen (90 N kg ha^−1^) is given only to those treatments that receive a total of 180 kg ha^−1^ of nitrogen. 

The above-ground parts were harvested, and all biomass from each field was weighed. The green biomass has two samples. The first sample was taken from chemical analysis before drying at 60 °C. The second sample was dried 105 °C. 

*Assessment of weediness.* Weediness in *Artemisia dubia* sward was evaluated in field experiments. The first weed assessment was carried out after spring fertilization in 2019 on May 13. In each variant, weed plantlet counts and the regrowth of *A. dubia* were calculated in two places per plot using a 0.25 m^2^ arc. The determined number of weeds and *Artemisia dubia* Wall plants is recalculated in units m^2^.

### 4.2. Laboratory Experiments

*Effects of Artemisia dubia on crops and weeds*. The effect of *Artemisia dubia* extracts on germination, seedling, and root length of spring wheat (*Triticume aestivum*), pea (*Pisum sativum*), rapeseed (*Brassica Napus*), common dandelion (*Taraxacum officinale*), and fat hen (*Chenopodium album*) was investigated.

*Obtaining the extract.* A control treatment of *Artemisia dubia* Wall, which was not fertilized, was selected to obtain the extract. After harvesting, *A. dubia* was chopped up to 5 cm and dried at room temperature. The dried chopped biomass of *A. dubia* was crushed in a grinder and weighed 10 g. The weighted mass was poured with 100 mL of distilled water and kept at 60 °C for 1 h in an ultrasonic water bath [38]. Different concentrations of 1:10 (weed biomass: distilled water), 1:50, 1:250, 1:1250, and 1:6250 were prepared from the obtained aqueous extract by dilution.

*Germination of seeds*. The seeds of the weeds and cultivated plants were germinated in a Petri dish on filter paper, 30 units each, filled with 5 mL of distilled water or extracts of different concentrations. The seeds of the plant were kept for 7 days in a controlled climate chamber, Climacell CLC-707-TV (Planegg, Germany), at a temperature of 24 °C and 65% humidity. After germination, the number of germinated weeds and cultivated plants was determined, and the length of the sprouts and roots was measured. The obtained results are compared with the results obtained in distilled water (control).

*Determination of dry matter.* The samples are dried in a drying chamber (Thermo Scientifi Heratherm OGS60 General Lab oven—Gravity, Frankfurt, Germany) at a temperature of 105 °C to a constant mass. The mass of the dry matter is calculated.

*Preparation of plant extracts for total polyphenolic compound (TPC) determination.* Approximately 0.5 g of raw air-dried *A. dubia* plant material was soaked with 10 mL of 75% methanol solution. The test tube with plant material and the solvent were thoroughly mixed using a vortex (IKA MS3, Wilmington, NC, USA) and treated in an ultrasonic bath for 60 min at ambient temperature. Then, the extracts were centrifuged at 1500 rpm (Rotanta 460, Hettich, Canada, USA) and filtered using a paper filter at 90 g/m^2^. Prepared extracts were stored at +4 °C until analysis.

*Total phenolic compound measurements.* The TPC was determined by the spectrophotometric method at a wavelength of 760 nm, following the standard procedure as described earlier [39]. Test samples were prepared by mixing 0.1 mL of the extract with 2.5 mL of bidistilled water, 0.1 mL of Folin–Ciocalteu reagent (Sigma-Aldrich, Darmstadt, Germany), and 0.5 mL of a 20% Na_2_CO_3_ solution. A blank sample was prepared following the same procedure; instead of extracting, solvent bidistilled water was used. The resulting solutions were incubated in the dark for 30 min. After that, their absorbance was measured using a UV-Vis spectrophotometer (Shimadzu, Kyoto, Japan). Thereafter, the concentration of TPC was calculated using the linear equation of the rutin calibration curve, y = 0.6454x + 0.1087, and the results are expressed as rutin equivalent mg g^−1^ of A. Dubia plants.

*Statistical analysis.* A three-way ANOVA analysis of variance was used for the statistical analysis of biomass productivity using R studio version 3.4.1 software. Data for other analyses were processed with the computer program SAS 7.1. Arithmetic means and standard deviations were calculated. Tukey’s test was used to determine significant differences. Statistical differences were made at the 0.05 significant level.

## 5. Conclusions

In summary, the multi-year yield results of *A. dubia* Wall show a multi-year yield potential. The highest biomass of the perennial plant was 18.3 t ha^−1^ in 2020 using N 180 kg ha^−1^ at 5000 per ha plants. A perennial plant species that will produce a considerable amount of biomass is also characterized by biologically active compounds and has an allelopathic effect. *A. dubia* Wall has potential allelopathic activity, and the aqueous extract of its biomass showed the inhibition of *T. Officinale* and *B. napus* seed germination. This study shows that *A. dubia* can be used as a bioherbicide to control different weeds. Therefore, extracts of this plant or the whole biomass of the plant may be useful for weed control in some crops. *A. dubia* biomass used like mulch can help farmers reduce their reliance on chemical herbicides when developing sustainable agricultural systems. Nevertheless, it is necessary to evaluate the biomass activity of *A. dubia* when used for weed control in field conditions.

## Figures and Tables

**Figure 1 plants-12-03750-f001:**
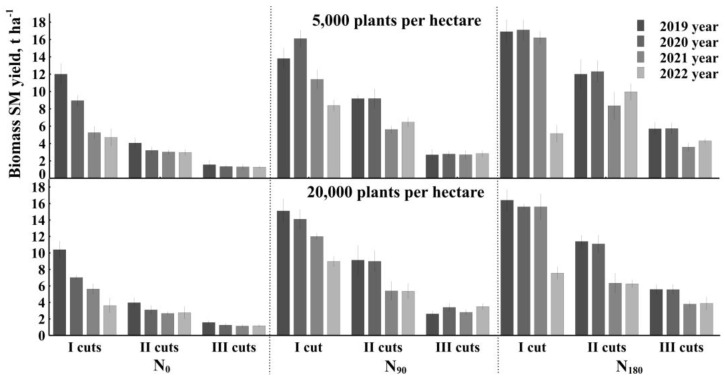
Four years of *Artemisia dubia* Wall productivity.

**Figure 2 plants-12-03750-f002:**
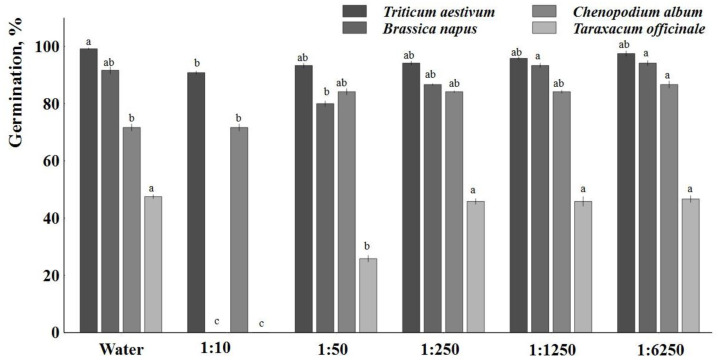
Allelopathic effect of *Artemisia dubia* Wall on the seed germination of crop plants and weeds. Note: Differences are significant at the 95% probability level if the means of the compared treatments do not have the same letter.

**Figure 3 plants-12-03750-f003:**
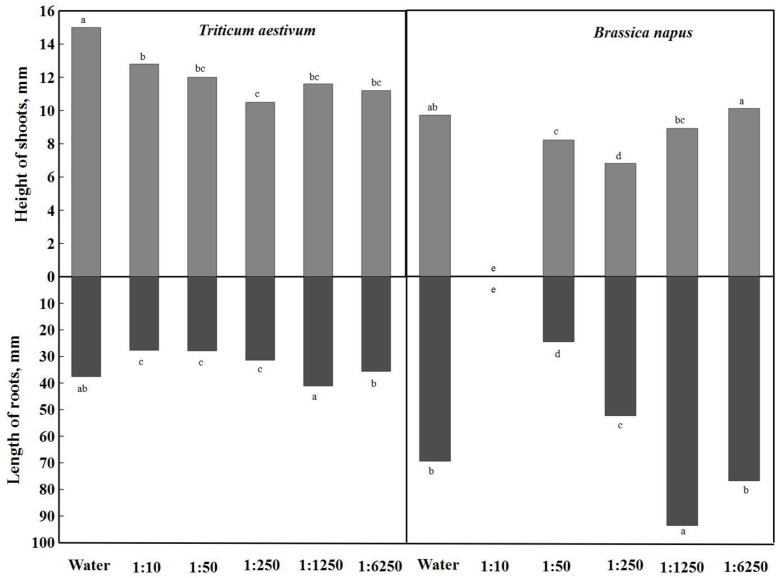
Effect of different concentrations of aqueous extracts of *A. dubia* on the shoots and roots of *T. astivum* and *B. napus*. Note: Differences are significant at the 95% probability level if the means of the compared treatments do not have the same letter.

**Figure 4 plants-12-03750-f004:**
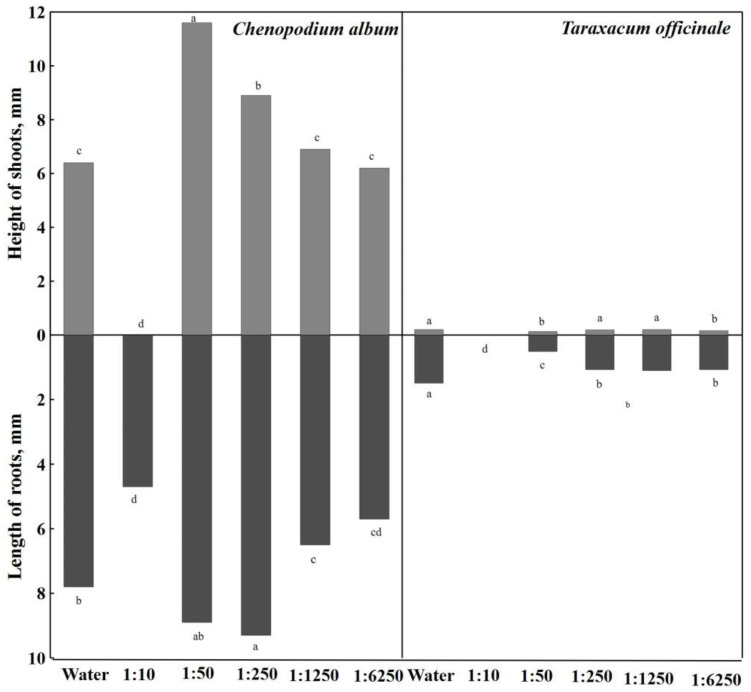
Effect of different concentrations of aqueous extracts of *A. dubia* on the shoots and roots of *C. album* and *T. officinale*. Note: Differences are significant at the 95% probability level if the means of the compared treatments do not have the same letter.

**Figure 5 plants-12-03750-f005:**
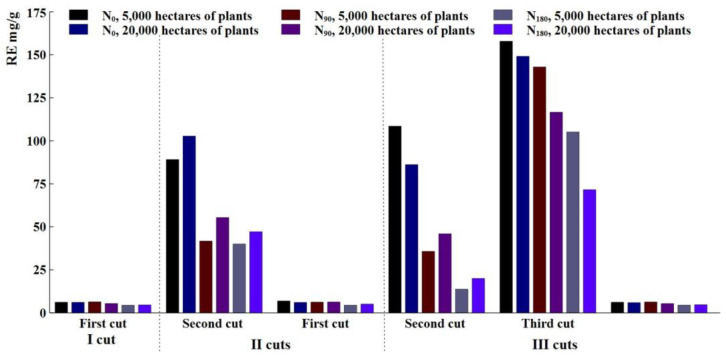
Amounts of phenolic compounds.

**Figure 6 plants-12-03750-f006:**
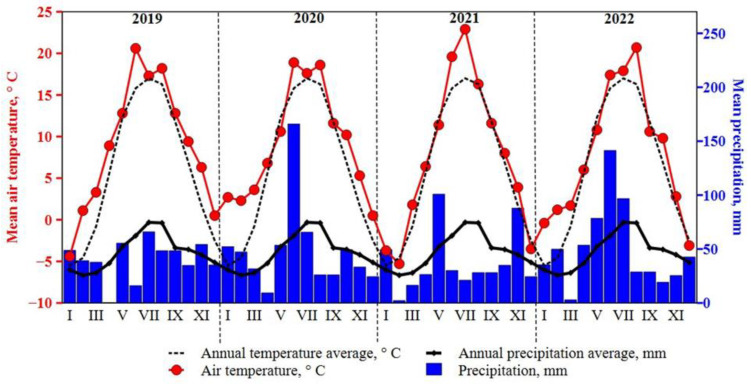
Meteorological conditions during experimental years 2019–2022.

**Table 1 plants-12-03750-t001:** Three factors for growing Artemisia Dubia Wall.

Factor A Densities	Factor B Fertilizers	Factor C Harvesting in the Growing Seasons
5000 plants per hectare	Control N_0_	1 cut
20,000 plants per hectare		
5000 plants per hectare	Ammonium nitrate N_90_	2 cuts
20,000 plants per hectare		
5000 plants per hectare	Ammonium nitrate N_180_	3 cuts
20,000 plants per hectare		

## Data Availability

Not applicable.

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
