# Peer review of "Potential of Artemisia dubia Wall Biomass for Natural Crop Protection"

_plants, 2023, doi:10.3390/plants12213750_

Round 1

Reviewer 1 Report

Comments and Suggestions for Authors

The paper is presented in a good way with clear indications of significant results and good visual representations. Is there any correlation between phenolic compound accumulation, biomass, and allelopathic effects? 

Reviewer 2 Report

Comments and Suggestions for Authors

Weeds reduce crop yield, increase the cost of production, degrade the quality of harvested product and reduce the quality of market product. Allelopathy has shown both inhibitory and stimulatory roles in plant processes such as on seed germination, overall growth, development, reproduction, disease/weed management, cell division, or biosynthesis of photosynthetic pigments of other plants by releasing some allelochemicals, mainly secondary metabolites. 

In the manuscript submitted by Baksinskaite et al. the authors analyzed the productivity of Artemisia dubia Wall and assessed the allelopathic effect of an aqueous biomass extract on crop and weed seeds.

 Detailed comments:

1.      There are no explicit research hypotheses in the manuscript.

2.      Lines 97-104: More emphasis should be placed on the importance of the results in terms of their use.

3.      Figure 1. Figures must be auto-explicative. In the description of the Y axis, add information whether it is dry or fresh mass?

4.      Lines 348. In the material and methods chapter, add information about the method of nitrogen fertilization. Was it a single dose (90 kg ha-1 and 180 kg ha-1) or divided, fertilization date, etc. Was the nitrogen fertilization only a result of the chemical composition of the soil?

5.      Line 370: Two densities were used in the experiment: 5,000 plants per hectare and 20,000 plants per hectare. Without conducting an experiment, it can be assumed that 20,000 plants per hectare is not the optimal density (over-densified cultivation). It was possible to check, for example, 12,000 plants per ha (as a third and intermediate variant). What was taken into account when selecting the density?

6.      Line 396: Please add information where this device (Climacell CLC-707-TV) was manufactured (country, city)?

7.      Lines 425-436: Conclusions: Specifically how and where these results can be used. Specify which crops (wheat?, spring rapeseed?). Furthermore, please consider that the conclusion is intended to help the reader understand why your research should matter to them.

 Technical notes:

8.      Figure 1. Zoom in on the descriptions of the X and Y axes.

9.      Figure 2. Zoom in on the descriptions of the X and Y axes. Enlarge legend.

10.  Figure 3 and 4. Zoom in on the descriptions of the X and Y axes.

11.  Figure 5. Please improve the quality of the figure. It is illegible in this shape. I suggest using different colors instead of shades of gray.

12.  Line 284: It is (2017). Should be: [26]

13.  Please check the Latin names throughout the text (e.g. lines 384, 431)

14.  Reference Lines 449-543: References  not prepared in accordance with the requirements journal Plants.

Author 1, A.B.; Author 2, C.D. Title of the article. Abbreviated Journal Name YearVolume, page range.

Round 2

Reviewer 2 Report

Comments and Suggestions for Authors

The authors addressed most of the problems. The manuscript is suitable for publication.